# Competitive advantages of organizational project management maturity: A quantitative descriptive study in Australia

**Garry Huang**[1], **Shwn-Meei Lee**[2], **Daniel L. Clinciu**[3]*

**1** College of Management, Taipei Medical University, Taipei, Taiwan, **2** Department of Applied English, Hsiuping University of Science and Technology, Taichung, Taiwan, **3** College of Management, National Chin-Yi University of Technology, Taichung, Taiwan

* celdan99@gmail.com

**Data Availability Statement:** All relevant data containing the minimal data set are within the paper and its Supporting Information files.

**Funding:** The authors received no specific funding for this work.

## Abstract

The purpose of this study is to examine whether organizational project management maturity in the project management consultancy industry offers any competitive advantages to a business when it tenders for contracts. We collected 150 responses from both members and former members of professional Australian project management associations in order to examine and understand any potential effects of project management maturity on management and organizations. The statistical software SPSS was used to analyze the data collected with the confidence interval (alpha) set at 5%. The results of this study reveal that organizational project management maturity has an effect on competitive advantages as noted by the project managers ($p < .0001$; 99.99%; $H0$ –rejected). The study also shows that the perception of competitive advantages for organizational project management maturity is based on the level of maturity reached. It also reveals that an organization is winning contracts/jobs due to various other competencies, particularly soft skills such as client relationships, stakeholder management, communications skills, and modes of client engagement.

## Introduction

Project management is widely used in the implementation of new programs and system changes seen in information technology projects, building projects, automotive products, airplanes, defense systems and many other areas [1–3]. Studies show that organizations having a standardized method of project management delivery increase their chances of delivering projects in accordance with plans [4, 5] and ultimately to the client's satisfaction.

Since adopting its modern form in the 1950s [6–9], project management has been an acquired skill, with various certifications and competency guidelines available to validate an individual's capability/competency. Such an example is the PRojects IN Controlled Environments (PRINCE2), a structured project management method and practitioner certification program [10, 11]. PRINCE2 emphasizes dividing projects into manageable and controllable stages. It is adopted in many countries worldwide, including the UK, Western European

**Competing interests:** The authors have declared that no competing interests exist.

countries, and Australia, with training available in many languages. There are many project management maturity models available for project managers to use (Appendix A, S1 File), nevertheless, they all emphasize that project managers need to be proficient in their own competency to achieve maturity. There are also general maturity models and methods that can further aid project managers in achieving competitive advantages (Appendix B, S1 File).

This study selected the Portfolio Programme and Project Management Maturity (P3M3) model, derived from PRINCE2 and developed in the United Kingdom, a country having a high degree of similarity with Australia. In addition, there are certifications available such as the Registered Project Management (RegPM) by the Australian Institute of Project Management and the Project Management Professional (PMP) provided by the Project Management Institute (PMI). PMI serves more than five million professionals including over 680,000 members in 217 countries and territories around the world, with 304 chapters and 14,000 volunteers serving local members in over 180 countries. Notably, the International Project Management Association (IPMA) is a federation of around 70-member associations across the globe which provides qualification standards for individuals working in project, programme and portfolio management [12]. The IPMA competency-based Four-Level Certification System for programme and project managers world-renowned for its quality and uniqueness. It comprises the following:

- Stage 1: Application, CV for initial certification, and Self-assessment.

- Stage 2: Examination.

- Stage 3: Final evaluation, Decision.

- Stage 4: Certification, Feedback, and Archiving.

This study, therefore, refers to competitiveness as the ability of project management organizations to secure contracts as they submit tenders.

## Theoretical background

Studies by Kathawala, Elmunti, and Toepp (1991); Barber (2004), and Milosevic and Patanakul (2005) revealed that organizations with standardized project management practices, in comparison to individual skill sets experience, improved overall organizational project management performance when assessed by the number of projects that meet the purpose, time and cost emphasized by the project plan [4, 13, 14]. Project management skills are shifting from an individual capability to an organizational competency as seen through various organizational project management maturity models [15]. Such skills encompass the soft skills of project managers which can play an important role in the success of a project by providing competitive advantages through better planning, execution, and evaluation of project progress. Soft skills are highly significant in providing various competitive advantages, especially when dealing with complex projects [16].

Among the skills set, the highly influential skills of project managers on project success include communication skills, team-building skills, and problem-solving skills. Current business environments encounter multiple challenges as well as opportunities in a volatile market environment. This situation requires constant changes within organizations and leaders' behavior. Studies on communication skills in the construction industry reveal that this particular soft skill can contribute as much as 78.3% variation in project success [17]. Another quantitive study showed that project managers' emotional intelligence, their team members' trust in them and job satisfaction also greatly impact the success of a project [18]. The shift in managing human capital with increasing complexity has developed project management from a

purely technical discipline to being viewed as a specialized field of management. Within it, leadership and interpersonal skills play a central role in future project enterprises and their success. Thus, current studies strongly suggest that project management interpersonal transferrable skills are the ones that will be most highly sought after currently and in the future [19].

## Problem statement

The specific problem this study aims to address is the lack of understanding of organizational project management maturity as defined by various maturity models and its effects on the perceived competitive advantages. Properly identifying the organizational project management maturity and its effects on the competitive advantages of an organization enable stakeholders to be better informed on both the strengths and weaknesses of the organization, thus, enabling the organization to continue evolving and developing. In this research, competitiveness refers to the ability of project management organizations to secure contracts when they submit a tender. This study examined the effects of organizational project management maturity on competitive advantages for Australian-based organizations.

## Methods

### Study participants

The participants were project managers, either current or former members of a professional project management association such as the Australian Institute of Project Management, the Project Management Institute, and the International Project Management Association. The study was approved by the Bellberry Human Research Ethics Committee (TGA HREC Code: EC00372).

A quantitative descriptive method was used to analyze the effects of organizational project management maturity level on competitive advantages for Australian-based project management consultation organizations; 130 project managers were surveyed. A quantitative method can interpret large amounts of data [20], and was therefore ideal to validate the specific research questions in this study. Furthermore, quantitative research allows for the testing of hypotheses from large amounts of data through precise measurement [21]. The main focus of this study was to collect empirical evidence of the experiences of the organizational project managers. Vogt et al., (2012) along with Ingham-Broomfield (2014) state that quantitative research aims at explaining and predicting solutions to a problem that could be generalized to other persons or places; additionally, there must be a clear objective reality that can be measured and quantified [22, 23]. While other research methods such as a qualitative study may also be a valid method, only a quantitative method can use statistics to interpret the large amounts of data [20].

### Design

Two components are the main focus of this study; an organizational project management maturity model self-assessment tool and a questionnaire employing a five-point Likert-type survey used in examining the perceptions of the project managers on organizational project management maturity level and its connection to competitive advantages. The purpose of such design is for allowing participants to document their perspectives on whether organizational project management maturity offers any competitive advantages when tendering for contracts. Participants were asked to fill and acceptance form (Appendix C, S1 File) to examine and complete a survey covering two main components. The first component was the Portfolio Programme Project Management Maturity Model (P3M3®) self-assessment tool, which focused on project management maturity (https://www.axelos.com/for-organizations/p3m3). The

second one, a five-point Likert-type survey required the participants to describe their attitudes, perceptions, and perspectives on the organizational competitive advantages associated with organizational project management maturity.

The collection and reporting the quantifiable descriptive data based on the experiences of the project managers on competitive advantages was central to this study. The quantitative descriptive design selected here enabled a systematic analysis of organizations across multiple levels suggesting that multiple factors may contribute to the overall perception of the surveyed individual [24], and is able to provide repeatable and statistically significant data to the variables [25]; for example, the perceived connection between organizational project management maturity and the competitive advantages outlined in this study. Therefore, by collecting the quantitative descriptive data of the project managers we aim to obtain a better understanding of a possible connection between organizational project management maturity as defined by various maturity models and the competitive advantages experienced by participants.

## Sampling

A sample size target of 125 is recommended by Cooper and Schindler (2012); as the population size approaches 125, a reasonable statistical estimate could be achieved from the standard deviation and is, therefore, a more meaningful result [21]. Therefore, for the purpose of this study, a sample size target of 125 valid responses was the target.

A convenience sampling method was used in this study; this is an appropriate method as potential subjects can choose either to participate or not in the study and is also an effective method of choice due to the ease of collecting sample data, its relatively low cost of collection [26] as well as allowing for the study to be completed in a shorter timeframe. However, a convenience sampling method could have a "coverage error" as described by [27].

## Research question

The main research question that this study is focused on is as follows: What is the effect of organizational project management maturity on the perceived competitive advantages as observed by the project managers that are current and former members of various professional project management associations?

This main question is further detailed and stratified into a 20-question survey.

## Field test

Three experts were invited to review the researcher developed survey questions and provide feedback on the appropriateness of the questions. Each expert has at least 20 years of project management experience and is either a current or former member with the Australian Institute of Project Management. Each expert responded accordingly and provided their review on the survey questions.

## Data analysis

A Likert-type survey consisting of 20 questions (P3M3/Appendix D, S1 File and Table 2) was administered to participants, with questions 10–20 (Table 2) focused on determining the organizational project management maturity level. The questions were based on the P3M3® project management self-assessment originally developed by the Office of Government Commerce. Questions 10–20 (Table 2) emphasized the experiences of the participants on competitive advantages provided by the organizational project management maturity and were developed by the researchers. The level of organizational project management maturity was

measured by the P3M3® project management self-assessment. The competitive advantages of the organization cannot be directly measured and are also considered as a latent variable [28]; that is, a single variable such as organizational project management maturity cannot be identified as the sole source of competitive advantages as there are many other factors involved. This claim was also supported by Agresti and Kateri (2014) in that a standard latent model treats the variables observed (level of organizational project management maturity) and latent variable (perceived competitive advantages) as a nominal scale, omitting any ordering that may exist [29]. Consequently, the addition or any other form of data manipulation is not possible based on findings by Field (2013) and Agresti and Kateri (2014). Therefore, the respondents were required to answer questions based on their experiences in the second component of the survey on whether or not they believe that organizational project management maturity was in fact one source of competitive advantages that they can rely on when submitting tenders.

The data collected by this study was then analyzed through the statistical software SPSS version 24. The confidence interval (alpha) was set at 5% and alpha inflation was auto-corrected by the statistical software automatically. As competitive advantages cannot be directly measured and are considered as a latent variable [28], then a factor analysis is an appropriate method for statistical analysis. The specific SPSS functions used included the"reliability analysis" in scale as well as the "factor analysis" in dimension reduction. For the hypotheses, the eigenvalues were determined by the number of survey questions and the computations done by SPSS [28]. Mathur, Jugdev, and Fung (2013) further suggested that by examining Cronbach's alpha, it is possible to gain an appreciation of how well the question relates to a unidimensional latent construct, and in this study, the perception of competitive advantages [30].

An eigenvalue of greater than one suggests a significant relationship with the corresponding factor [28], therefore, revealing that the participants agree or highly agree with the corresponding survey question (statement) and Nunnally (1978) suggested that a new research topic with a Cronbach's alpha of 0.5 would be adequate to determine the reliability of the study [31].

## Grouping of data

Of the expected responses, the data were separated into two main groups and contrasted against each group. One group consisted of the survey responses with an organizational project management maturity level ranging between one to three and the other group ranged between four and five. Each group was entered into SPSS for factor analysis to determine its eigenvalues and the Cronbach' alpha. The results were contrasted in order to determine any similarities.

The purpose for establishing a baseline is to be able to group the responses by the survey participants into similar categories. This is to ensure that a reasonable comparison between the survey responses can be achieved. Without the grouping, it is possible that only a very small amount of organizations would fall into level one and level five maturities and therefore making it impossible to contrast.

## Results

A total of 150 participants initially participated in this study, however, 20 participants were excluded after failing to agree to terms and conditions, resulting in a total number of 130 participants (n = 130). SPSS further excluded 21 data sets due to incomplete responses resulting in a total of 109 valid responses (Table 1, S1 File). The average number of years of work experience for the participants was 13.73 years. From the study's protocols, the Cronbach alpha was calculated at 0.615 (Table 2, S1 File) for the 11 components used (Table 1) which determined this study has a sufficiently reliable result (0.5 minimum used as the acceptance criteria).

**Table 1. Total variance among the components of this study.**

| Component | Initial Eigenvalues | |
|:---:|:---:|:---:|
| | Total | % of Variance |
| 1 | 6.413 | 58.299 |
| 2 | 1.121 | 10.187 |
| 3 | .757 | 6.879 |
| 4 | .707 | 6.424 |
| 5 | .478 | 4.346 |
| 6 | .425 | 3.867 |
| 7 | .302 | 2.748 |
| 8 | .270 | 2.453 |
| 9 | .206 | 1.873 |
| 10 | .191 | 1.734 |
| 11 | .131 | 1.192 |

*Note.* Number of Components identified = 2

The total variance calculated by SPSS identified two components with an eigenvalue greater than one using survey questions 10 through 20 (Table 2). The two components identified relevant survey questions to be used in determining maturity levels 1–5, thus, removing questions 12 and 20 (Table 3). Based on the self-assessment P3M3 questions, maturity levels were grouped into two sections: maturity levels 1–3 and 4–5.

These two components were able to explain 68.5% of the total variance in this study with each component being able to explain 58.3% and 10.2% of the variance respectively. Component two as identified by the Eigenvalue calculation suggested that question 16 shows a high degree of variance to the overall calculations (Tables 2 and 3).

## Independent samples T-test results

The independent sample T-test as calculated by SPSS revealed a p-value of <0.0001 (99.99% confidence, Table 3, S1 File; thus rejecting the null hypothesis ($H0$ –Organizational project

**Table 2. Survey questions used for determining components' matrix outcome.**

| Question# | Survey question | Component 1 | Component 2 |
|:---:|:---|:---:|:---:|
| 10 | Are your organization's project management Capabilities a contributing factor when bidding for contracts/jobs? | .892 | .112 |
| 11 | The organization has always invested in professional development of project managers as a source of competitive advantages | .851 | -.007 |
| 12 | The organization emphasizes the importance of client relationships over the importance of organizational project management maturity | -.549 | .399 |
| 13 | The organization only secures contracts/jobs if it charges a lower price than competitors | -.789 | -.038 |
| 14 | The organization can compete with its nearest competitor in all its organizational project management capabilities | .866 | .098 |
| 15 | The organization secures contracts/jobs based on its organizational project management maturity/capabilities | .879 | .053 |
| 16 | The organization is winning contracts/jobs due to other competencies (e.g. soft skills: relationships, management, communications) | -.106 | .868 |
| 17 | The organization often benchmarks its own organizational project management maturity for the purpose of continuous improvement (e.g. development, workshops, staff education) | .784 | .157 |
| 18 | The organization has created new business opportunities because of its organizational project management maturity | .869 | .072 |
| 19 | Do you agree with the statement: "Organizational project management maturity has given the organization a competitive edge?" | .897 | .072 |
| 20 | Do you agree with: "It is not about what you know, but who do you know" in the current market? | -.510 | .383 |

**Table 3. Survey questions and scores for possible competitive advantages in maturity levels 1 to 3 and 4 to 5.**

| Question# | Survey question | Score: (Levels 1–3) | (Levels 4–5) |
|---|---|---|---|
| 10 | Are your organization's project management Capabilities a contributing factor when bidding for contracts/jobs? | 3.24 | 4.46 |
| 11 | The organization has always invested in professional development of project managers as a source of competitive advantages | 2.53 | 3.86 |
| 13 | The organization only secures contracts/jobs if it charges a lower price than competitors | 2.95 (3.05[a]) | 1.78 (4.22[a]) |
| 14 | The organization can compete with its nearest competitor in all its organizational project management capabilities | 2.75 | 4.49 |
| 15 | The organization secures contracts/jobs based on its organizational project management maturity/capabilities | 2.85 | 4.14 |
| 16 | The organization is winning contracts/jobs due to other competencies (e.g. soft skills: relationships, management, communications) | 4.27 | 4.32 |
| 17 | The organization often benchmarks its own organizational project management maturity for the purpose of continuous improvement (e.g. development, workshops, staff education) | 2.74 | 3.83 |
| 18 | The organization has created new business opportunities because of its organizational project management maturity | 2.55 | 3.70 |
| 19 | Do you agree with the statement: "Organizational project management maturity has given the organization a competitive edge?" | 2.97 | 4.13 |

[a] The adjusted score

management maturity has no effect on competitive advantages as perceived by the project managers).

Maturity levels 1–3 represent 47 respondents, and Maturity levels 4–5 represent 62 (21 respondents were excluded). Based on the final question "Do you agree with the statement: Organizational Project Management Maturity has given the organization a competitive edge", organizations with a maturity level of 1–3, scored on average 2.97 (Table 4, S1 File) while those with a maturity level of 4–5 scored 4.13 (Table 5, S1 File, Table 3).

## Discussion

This study examined the effects of organizational project management maturity on competitive advantages for organizations based in Australia. SPSS identified two components with an eigenvalue greater than one, suggesting there are two factors with particular significance in this study (Table 1). The second component (question 16), based on SPSS calculations identified 10.2% of the variance (eigenvalue of 1.121). This component relates to the survey question which addresses the impact of soft skills on the ability to win contracts/jobs (Tables 3 and 6, S1 File). The score of 4.30 (agree to strongly agree), suggests that the surveyed project managers considered this feature an important aspect of competitive advantages. Although previous studies found soft skills to be of particular importance in a variety of fields [17–19], they did not investigate whether such skills could be used as competitive advantages for securing or winning contracts.

This study revealed that the perception of competitive advantages based on organizational project management maturity is based on the level of maturity (Table 3). Project managers in organizations having lower levels of maturity do not perceive various competitive advantages, whereas, project managers in organizations with higher levels of maturity do. Also, it is possible to replicate this study using different maturity models besides the P3M3® used in this study such as the Portfolio Management Maturity Model (PfM3) and the Programme Management Maturity Model (PgM3). This study did not, however, factor in a weighted component with each of the survey questions, and any future research can potentially benefit from a weighted score for different survey questions.

The results of this study demonstrated there is a higher perception of competitive advantages in organizations with a higher level of organizational project management maturity. One

interesting aspect to note was that all of the participants were perceived to possess additional competitive advantages. However, the definition of competitive advantage is a resource or strategic asset that is valuable, rare, inimitable, and involves organizational focus [32], and since it is possible for any organization to reach the same level of organizational project management maturity, it is not rare and inimitable. Therefore, future research should investigate through weighted component questions based on the outcome of this study to determine other possible sources of competitive advantages.

One specific potential area for future research is to conduct this research in a different country, which is a limitation of this study. Asian countries rely more on human interactions rather than the technical skills of an individual, occasionally known as Hofstede's cultural dimensions [33]. Therefore, by repeating this study in a different country, particularly an Asian country, better observation of the impact of the second significant component in this study and its relationship to competitive advantages may occur. Another potential study, using a qualitative phenomenological method and design that could gather the lived experiences of project managers based in another country with significant cultural differences incorporating organizational project management maturity when developing programs may be explored. Rather than looking at an entire country, focusing on one particular city's project managers in a focus group design, might provide another perception of the interface between project management maturity and competitive advantages, especially with respect to the soft skills of project management.

## Conclusion

Whether organizational project management maturity can be relied upon as the single source of competitive advantages is not a conclusion of this study. Based on the independent sample T-test as calculated by SPSS, a p-value of $<0.0001$ (99.99% confidence) was obtained which rejected the null hypothesis (*N*0: Organizational project management maturity has no effect on competitive advantages as perceived by the project managers). This suggests that there is a relationship between organizational project management maturity and the perceived competitive advantages of the surveyed project managers. Furthermore, this study confirmed that the soft skills of project management such as client relationships (with an overall average of 4.30), stakeholder management, and communications abilities are important factors for winning jobs/contracts across all levels of organizational project management maturity. By properly identifying the organizational project management maturity and its effects on the competitive advantages of an organization, the stakeholders may be more aware of the strengths and weaknesses of the organization, thus, prompting the organization to continue developing and evolving.

## Supporting information

**S1 File.**
(DOCX)

## Author Contributions

**Conceptualization:** Garry Huang, Daniel L. Clinciu.

**Data curation:** Garry Huang.

**Formal analysis:** Shwn-Meei Lee, Daniel L. Clinciu.

**Investigation:** Garry Huang, Shwn-Meei Lee.

**Methodology:** Daniel L. Clinciu.

**Project administration:** Shwn-Meei Lee.

**Software:** Shwn-Meei Lee, Daniel L. Clinciu.

**Supervision:** Garry Huang, Shwn-Meei Lee.

**Validation:** Shwn-Meei Lee, Daniel L. Clinciu.

**Visualization:** Daniel L. Clinciu.

**Writing – original draft:** Garry Huang, Daniel L. Clinciu.

**Writing – review & editing:** Daniel L. Clinciu.

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
