## [Decision Letter · Decision Letter 0]

2 Oct 2022

PONE-D-22-12656Competitive Advantages of Organizational Project Management Maturity: A Quantitative Descriptive Study in AustraliaPLOS ONE

Dear Dr. clinciu,

Thank you for submitting your manuscript to PLOS ONE. After careful consideration, we feel that it has merit but does not fully meet PLOS ONE’s publication criteria as it currently stands. Therefore, we invite you to submit a revised version of the manuscript that addresses the points raised during the review process.

Please note that we have only been able to secure a single reviewer to assess your manuscript. We are issuing a decision on your manuscript at this point to prevent further delays in the evaluation of your manuscript. Please be aware that the editor who handles your revised manuscript might find it necessary to invite additional reviewers to assess this work once the revised manuscript is submitted. However, we will aim to proceed on the basis of this single review if possible. The reviewer has identified several opportunities to clarify aspects of the study design and to provide additional contextualisation. Please respond carefully to each of the points they have raised when preparing your revisions.

We look forward to receiving your revised manuscript.

Kind regards,

Jamie Males

Editorial Office

PLOS ONE

Journal Requirements:

"No funding was received for this study"

Reviewers' comments:

Reviewer's Responses to Questions

**Comments to the Author**

1. Is the manuscript technically sound, and do the data support the conclusions?

Reviewer #1: Yes

2. Has the statistical analysis been performed appropriately and rigorously? 

Reviewer #1: Yes

3. Have the authors made all data underlying the findings in their manuscript fully available?

Reviewer #1: Yes

4. Is the manuscript presented in an intelligible fashion and written in standard English?

Reviewer #1: Yes

5. Review Comments to the Author

Reviewer #1: Formal comments:

1. There is not clear which of the resources (Itim Interantiona and Itim International, 2010) on page 15 and in References is in right form (I suppose the second one). It is necessary to correct it.

Content comments:

1. Introduction – 1st paragraph - project management is discipline used not only in “… information technology projects, building projects, new automotive products, airplanes, or weapons systems ...”. I recommend to modify it.

2. Introduction – 2nd paragraph – The skills of project managers could be certified. The mentioned certification from Australia is regional certification, I recommend to mention certifications used worldwide, not only Project Management Institute certification, but also PRINCE2 and International Project Management Association standards certifications.

3. Results – it will be useful to present in the article also description of answers of respondents. If they full filled self-assessment P3M3 questions it could be possible to present what is the level of project management maturity of respondent’s organizations. This information could give the overview about the maturity of project management. It could be important information, in case the maturity is mainly on low level (P3M3 has got 5 levels – from the lowest “awareness of process” to the highest “optimized process”) it could be the reason of the final statement: „organizational project management maturity has no effect on competitive advantages “. There would be very interesting the comparison of high-level project maturity organizations and low-level project maturity organizations – is there for both groups still the result same, or the level of project maturity could impact the relationship between organizational project management maturity and competitive advantages?

4. References – there is possible to find many references but mainly to methodology of the research, to the main topic – project management maturity and competitive advantages – is not presented sufficient literature review.

5. Supplementary material – there is presented the Project Management Maturity Model (P3M3®) self-assessment, but this project management maturity model is described in the literature, so it is sufficient just to offer to the readers the link to description of this model. The questions 10-20 are presented two times – in table and in supplementary material, it is redundant.

Summary:

Authors would like to examined the effects of organizational project management maturity on competitive advantages. The methodology, problem statement and design of the analysis are specified very clearly and deeply. But the survey is very limited by insufficient literature review. The methodology and specification of research are based on recommendation from literature, but there is not presented the theoretical background of the project management maturity and competitive advantages.

The project management maturity is possible to measure by project management maturity models. There are, based on the literature review, available more than 50 project management maturity models. Why the P3M3 by Axelos based on PRINCE2 methodology has been used for the analysis? Is this methodology preferred in Australia? There are other models, like OPM3 (based on Project Management Institute standard), IPMA Delta (based on International Project Management Association standards), Kerner Project Management Maturity Model (by Kezner, 2019, 2005), Project Management Solutions´ Project Management Maturity Model (by Crawford, 2021), Project Management Process Maturity by Kwak and Ibbs (2002). Project FRAMEWORKTM Project Management Maturity Model (by Iqbal 2016), all based on PMI standard and many others.

Why the model P3M3 has been chosen? There are available other self-assessment models. Is it based on the statement, that PRINCE2 standard is the most preferable in Australia? If not, there is not presented the relevant explanation the P3M3 model has been chosen for this survey.

It is same with the competitive advantages, the second part of the survey is focused on project managers´ attitude to relationship of project management maturity and competitive advantages, but there is not literature review, which would specify in detail how the competitive advantages could be specified. There is not mentioned if these relationships has been analysed before in some other studies, or if they are focused just on comparison of success and project management maturity.

International Project Management Association. (2016). Reference model for IPMA Delta. http://www.ipma.world/certification/certify-organisations/delta-reference-model/

Iqbal, S. (2016). Organizational Maturity – Managing Programs Better. In. "Program Management: A Life Cycle Approach", CRC Press, edited by Ginger Levin 584 pp.

Kerzner, H. (2019). Using the Project Management Maturity Model, Third Edition: Strategic Planning for Project Management. John Wiley &Sons, Inc., doi: 10.1002/9781119559078.

Kerzner, H. (2005). Using the project management maturity model. Hobokon, New Jersey: John Wiley& Sons.

Kwak, Y. H.; Ibbs, C. W. (2002). Project management process maturity (PM)2 model. Journal of Management in Engineering, 18, 150-155.

Project Management Institute (2003). Organizational Project Management Maturity Model (OPM3). 1. ed. Newton Square: PMI, 195 pp. doi: 10.1201/9781420028942.axa

Crawford, J. K. (2021). Project Management Maturity Model (PM Solution Research). 4. ed., Auerbach Publications, 234 pp.

6. PLOS authors have the option to publish the peer review history of their article (what does this mean?). If published, this will include your full peer review and any attached files.

Reviewer #1: **Yes: **Jana Kostalova

---

## [Author Response · Author response to Decision Letter 0]

14 Oct 2022

Responses to Reviewers

We want to thank the reviewer for her time and valuable comments. Our editing and changes have been highlighted in red throughout the manuscript. 

1. Introduction – 1st paragraph - project management is discipline used not only in “… information technology projects, building projects, new automotive products, airplanes, or weapons systems ...”. I recommend to modify it.

1. We have edited parts of the abstract that may have been confusing and corrected the misspelled word and also its reference format (added the year in the references section).

2. Introduction – 2nd paragraph – The skills of project managers could be certified. The mentioned certification from Australia is regional certification, I recommend to mention certifications used worldwide, not only Project Management Institute certification, but also PRINCE2 and International Project Management Association standards certifications.

2. We have added a short description of PRINCE2 in the introduction. We include in our study an Appendix (Appendix A) which lists over 20 different maturity models which suggest that project managers need to be proficient in their own competency in order to achieve maturity. We hope this can satisfy the reviewer’s concern.

3. Results – it will be useful to present in the article also description of answers of respondents. If they full filled self-assessment P3M3 questions it could be possible to present what is the level of project management maturity of respondent’s organizations. This information could give the overview about the maturity of project management. It could be important information, in case the maturity is mainly on low level (P3M3 has got 5 levels – from the lowest “awareness of process” to the highest “optimized process”) it could be the reason of the final statement: „organizational project management maturity has no effect on competitive advantages “. There would be very interesting the comparison of high-level project maturity organizations and low-level project maturity organizations – is there for both groups still the result same, or the level of project maturity could impact the relationship between organizational project management maturity and competitive advantages?

3. We added additional details of the responses’ results in the Results section to clarify our findings and to answer the reviewer’s points. We also added a new table (Table 2) and modified Table 3 by adding the results of maturity groups in it: 1) maturity levels 1 – 3 and 2) maturity levels 4 – 5 in order to clarify our findings and make it easier for the reader to understand them. Please see the red highlights in the results section.

4. References – there is possible to find many references but mainly to methodology of the research, to the main topic – project management maturity and competitive advantages – is not presented sufficient literature review.

4. We have added more details in the introduction, and also references to support the main topic. Please see the red highlights in the introduction (paragraphs 2 and 3, and references).

5. Supplementary material – there is presented the Project Management Maturity Model (P3M3®) self-assessment, but this project management maturity model is described in the literature, so it is sufficient just to offer to the readers the link to description of this model. The questions 10-20 are presented two times – in table and in supplementary material, it is redundant.

5. We have removed the redundant material/information in the supplementary material and added necessary links, appendixes, and tables. Please see the new revised the tables in the manuscript.

---

## [Editor Report · Decision Letter 1]

9 Jan 2023

PONE-D-22-12656R1Competitive Advantages of Organizational Project Management Maturity: A Quantitative Descriptive Study in AustraliaPLOS ONE

Dear Dr. clinciu,

Thank you for submitting your manuscript to PLOS ONE. After careful consideration, we feel that it has merit but does not fully meet PLOS ONE’s publication criteria as it currently stands. Therefore, we invite you to submit a revised version of the manuscript that addresses the points raised during the review process.

We look forward to receiving your revised manuscript.

Kind regards,

Jana Košťálová, Ph.D.

Guest Editor

PLOS ONE

Journal Requirements:

Additional Editor Comments (if provided):

Authors accepted most of the comments of reviewer. The most import is including of PRINCE2 standard and the overview of project management maturity measured by P3M3 of respondents, which gives relevant overview on which level of project management in these organizations is and how relevant are their answers in questionnaire survey.

The is not extended very much the literature review, but there is appendix A, where the available maturity models are included.

The most important parts are improved, but there are still some formal and content comments:

Formal comments:

1. There is not the title of Table 1, please include it.

Content comments:

1. Introduction – 1st paragraph - project management is discipline used not only in “… information technology projects, building projects, new automotive products, airplanes, or weapons systems ...”. I recommend to modify it.

Insufficient formulation, I recommend to modify it in this way: … project management is used much widely i.e. IT projects, building projects….. and many other areas.

2. Introduction – 2nd paragraph – The skills of project managers could be certified. The mentioned certification from Australia is regional certification, I recommend to mention certifications used worldwide, not only Project Management Institute certification, but also PRINCE2 and International Project Management Association standards certifications.

The text is intended for readers across the world. So, the overview in the introduction has to reflect the situation across the world, not only in Australia. Generally, across the world PMI, PRINCE2 and IPMA international project management standards and certifications for project managers skills are used (despite the fact that IPMA does not have its branch in Australia). I recommend to mention in the Introduction also this third standard, to offer general independent overview of project management standards.

3. Appendix A – the title of the Appendix is not accurate. This is not a list of project management maturity models. There are included general project management maturity models (like Building Architecture Maturity Model, Capability Maturity Model Integration for Development, Capability Maturity Model Integration for Acquisition, Corrective Maintenance Maturity Model or EFQM Excellence Model). There are general management methods (like Balance Scorecard). Some of these models has been used for development of project management maturity model (like EFQM Excellence Model or CMMI model). These models and method are usable for evaluation of management maturity generally, not exactly for project management maturity.

There are mentioned international project management standards (like International Project Management Association and its ICB standard), but it is not project management maturity model. Based on this standard is available Reference model IPMA Delta (IPMA, 2016). Unfortunately, the IPMA Delta project management maturity model is not included in the list.

There is only 15 real project management maturity models int the appendix. It is necessary to specify it precisely. I recommend to modify table – stay there only real project management maturity models or divide it into two parts – general maturity models and methods and project management maturity models.

International Project Management Association. (2016). Reference model for IPMA Delta. http://www.ipma.world/certification/certify-organisations/delta-reference-model/
---

## [Author Response · Author response to Decision Letter 1]

9 Feb 2023

Format comments

We have provided a title for Table 1. Please see the title in red highlights (revised manuscript)

Content comments

We have modified the Introduction paragraph as per reviewer’s suggestion. Please see the red highlights in paragraph 1.

We have added more content as per reviewer’s suggestion. Thank you for the great example and please see red highlights

We have separated the appendix into two parts as suggested by the reviewer. We now have appendixes A (project management maturity models) and B (General Maturity Models and Methods).

---

## [Decision Letter · Decision Letter 2]

16 Feb 2023

PONE-D-22-12656R2Competitive Advantages of Organizational Project Management Maturity: A Quantitative Descriptive Study in AustraliaPLOS ONE

Dear Dr. clinciu,

Thank you for submitting your manuscript to PLOS ONE. After careful consideration, we feel that it has merit but does not fully meet PLOS ONE’s publication criteria as it currently stands. Therefore, we invite you to submit a revised version of the manuscript that addresses the points raised during the review process.

We look forward to receiving your revised manuscript.

Kind regards,

Jana Košťálová, Ph.D.

Guest Editor

PLOS ONE

Journal Requirements:

Reviewers' comments:

Reviewer's Responses to Questions

**Comments to the Author**

1. If the authors have adequately addressed your comments raised in a previous round of review and you feel that this manuscript is now acceptable for publication, you may indicate that here to bypass the “Comments to the Author” section, enter your conflict of interest statement in the “Confidential to Editor” section, and submit your "Accept" recommendation.

Reviewer #2: All comments have been addressed

2. Is the manuscript technically sound, and do the data support the conclusions?

Reviewer #2: Partly

3. Has the statistical analysis been performed appropriately and rigorously? 

Reviewer #2: Yes

4. Have the authors made all data underlying the findings in their manuscript fully available?

Reviewer #2: No

5. Is the manuscript presented in an intelligible fashion and written in standard English?

Reviewer #2: Yes

6. Review Comments to the Author

Reviewer #2: - There is not a cross reference No. 4 to the author name.

- There are some typing errors (e.g. “…Dorouin & Jugdev, 2014….. ”; correct is “…Drouin & Jugdev, 2014….. ”; – see Introduction, second paragraph).

- I recommend to add the chapter Theoretical Background (Literature Review) which evaluates the current state of the research topic on an international scale. – I recommend move the last paragraph from the Introduction chapter to this chapter and further expanding this chapter with other literary sources especially from the Web of Science or Scopus databases not older than 5 years.

- I recommend more in detail describe the methodology (i.e. the used statistical methods) which help to achieve the defined research aim.

7. PLOS authors have the option to publish the peer review history of their article (what does this mean?). If published, this will include your full peer review and any attached files.

Reviewer #2: No

---

## [Author Response · Author response to Decision Letter 2]

2 May 2023

We have addressed all of the concerns of the reviewer

We have removed the cross reference #4 in the title page. 

We have corrected the spelling error of Drouin (highlighted in red)

We have added a section in the Introduction titled Theoretical Background which includes references and/or literary sources within the last 5 years as per reviewer’s suggestion. Thank you for your details (please see the red highlights).

We have provided more details of the methodology as per reviewer’s suggestion (please see red highlights in the Methods section).

---

## [Editor Report · Decision Letter 3]

2 Jun 2023

Competitive Advantages of Organizational Project Management Maturity: A Quantitative Descriptive Study in Australia

PONE-D-22-12656R3

Dear Dr. clinciu,

We’re pleased to inform you that your manuscript has been judged scientifically suitable for publication and will be formally accepted for publication once it meets all outstanding technical requirements.

Kind regards,

Jana Košťálová, Ph.D.

Guest Editor

PLOS ONE

Additional Editor Comments (optional):

Dear authors,

thank you for accepting of the requirements and correction of the text based on them.

Best Regards

Guest Editor
---

## [Editor Report · Acceptance letter]

19 Jun 2023

PONE-D-22-12656R3 

Competitive Advantages of Organizational Project Management Maturity: A Quantitative Descriptive Study in Australia 

Dear Dr. Clinciu:

I'm pleased to inform you that your manuscript has been deemed suitable for publication in PLOS ONE. Congratulations! Your manuscript is now with our production department. 

Kind regards, 

on behalf of

Dr. Jana Košťálová 

Guest Editor

PLOS ONE